# The Effects of Salinity on the Survival, Growth, and Eco-Physiological Parameters of Juvenile Sea Urchin *Diadema setosum*

**DOI:** 10.3390/ani15162462

**Published:** 2025-08-21

**Authors:** Xuanliang Wang, Jieyu Zhang, Lei You, Yunyong Jin, Zhenhao Lin, Junhao Lin, Jinhui Wu, Zonghe Yu

**Affiliations:** 1College of Marine Sciences, South China Agricultural University, Guangzhou 510642, China; 19878885409@163.com (X.W.); jieyuzhang0214@163.com (J.Z.); 13155896195@163.com (L.Y.); 18067766476@163.com (Y.J.); 18338923923@163.com (Z.L.); 18929506095@163.com (J.L.); 2Agro-Tech Extension Center of Guangdong Province, Guangzhou 510520, China; 15715006151@163.com

**Keywords:** *Diadema setosum*, juvenile, salinity, survival, growth, eco-physiological parameters

## Abstract

*Diadema setosum* holds both economic and ecological significance. Economically, it is not only valued for its medicinal properties but also for its gonad, which can be processed into various seafood products, making it an important commercial sea urchin species. Ecologically, *D. setosum* plays a vital role in maintaining marine ecosystem balance by feeding on algae and organic debris, thereby facilitating nutrient cycling and energy flow within benthic ecosystems. However, overfishing and environmental changes have led to a decline in its population. Previous studies have shown that salinity significantly affects the reproduction and larval development of *D. setosum*, but its impact on juveniles remains unclear. In this study, we investigated the effects of salinity on the survival, growth, and eco-physiological parameters of juvenile *D. setosum*. Our results demonstrate that salinity has a substantial influence, with an optimal range of 32–36. This work provides valuable insights for the breeding and aquaculture of *D. setosum.*

## 1. Introduction

Sea urchins are prized as a premium marine delicacy, particularly valued for their nutrient-rich gonads containing high-quality protein, essential amino acids, and bioactive compounds such as polyunsaturated fatty acids (PUFAs) and β-carotene [1]. This commercial value drives significant global markets in China, Japan, and France [2,3,4], where premium specimens command prices up to EUR 30/kg wholesale and processed gonads reach EUR 1035/kg [5]. However, exploitation is constrained by biological limitations; fewer than 30 of ~1000 known species support commercial fisheries [6,7], and compounding this scarcity, wild populations face severe depletion from overfishing. Traditional harvest methods exhibit low profitability, rendering aquaculture—particularly sea-based systems with algal/mixed diets [5,8]—the critical solution for sustainable production.

*Diadema setosum* (Echinoidea: Diadematidae) constitutes an ecologically and economically critical Indo-West Pacific species, functioning as both a keystone herbivore structuring coral/rocky reef ecosystems through bioerosion-mediated nutrient cycling [4,7,9,10,11,12]. This dual ecological role sustains its commercial significance in ornamental and food industries [4,7,10,11,12,13,14,15,16]. Wild populations are facing escalating pressures from multiple anthropogenic and environmental stressors, including habitat degradation from tourism infrastructure [10], coral reef resource depletion [7], and climate-mediated pathogenic outbreaks. The latter is exemplified by the 2023 Vibrio-driven mass mortality along Türkiye’s Aegean coast where clinical symptoms (spine loss, mucoid layers) and laboratory confirmation established vibriosis as the primary etiology, with parasitic agents excluded. This mortality event corresponded spatiotemporally with population collapses from 150 to 2.3 ± 1.9 ind./100 m^2^, exacerbated by thermal anomalies (peak 26 °C) promoting *Vibrio* proliferation [17]. Concurrently, unsustainable harvesting intensifies population declines to meet rising market demand [4,10,15]. These synergistic drivers accelerate benthic macroinvertebrate community collapse and critically compromise ecological functions, necessitating the urgent development of climate-resilient captive breeding technologies for sustainable conservation.

Sea urchins exhibit exceptional osmoregulatory sensitivity owing to their distinctive anatomical adaptations. Their extensive seawater interface (via body surfaces, peristomial gills, and madreporite pores) confers heightened salinity sensitivity, likely due to direct osmotic exchange with the water vascular system [18]. Echinoderms primarily maintain osmotic balance through intracellular isotonic regulation and ion regulation [9,19,20,21,22]. Previous studies have documented species-specific salinity tolerances; the larvae of *Strongylocentrotus droebachiensis* demonstrate tolerance to salinities as low as 20 [9]. *Heliocidaris crassispina* exhibits developmental sensitivity [18], *Hemicentrotus pulcherrimus* achieves optimal fertilization (95–100%) at salinities of 27–35 [19], and *Strongylocentrotus intermedius* shows delayed embryogenesis under hyposaline conditions [21].

For *D. setosum*, research indicates that it exhibits optimal reproductive performance at a salinity of 31, with fertilization rates, larval survival, and growth all peaking at this level. By contrast, when salinity falls below 28 or exceeds 37, both reproductive capacity and larval development become significantly inhibited [23]. However, systematic investigations of salinity effects on juvenile stages remain lacking. This study comprehensively evaluates salinity effects on the survival, growth, and eco-physiological parameters of juvenile *D. setosum*, providing critical data to advance tropical sea urchin aquaculture technologies (see Appendix A).

## 2. Materials and Methods

### 2.1. Experimental Materials

On 14 September 2023, broodstock of *D. setosum* were collected from their natural habitat in Daya Bay, the South China Sea (22°34′ N, 114°33′ E). Captured individuals were transported to and temporarily maintained at the Shenzhen Experimental Base, South China Sea Fisheries Research Institute, Chinese Academy of Fishery Sciences. On 16 September 2023, larval rearing and nursery culture of experimental juveniles were initiated following a standardized protocol [24,25]. All experimental juveniles were derived from this batch and were maintained on a diet of benthic diatoms and fresh *Sargassum*.

The experiment was conducted in two phases, a winter trial starting on 12 January 2024 and a spring trial beginning on 15 April 2024. In the winter experiment, we utilized juvenile *D. setosum* with mean test diameters of 5.10 ± 0.70 mm and wet weights of 0.08 ± 0.01 g. In the spring experiment, the juveniles exhibited larger sizes, with test diameters measuring 7.50 ± 0.80 mm and wet weights averaging 0.24 ± 0.07 g.

### 2.2. Experimental Methods

The controlled laboratory experiment was conducted under natural photoperiod conditions using 18 independent 20 L tanks. A fully factorial design was employed with replicate seasonal trials (winter and spring) across six salinity treatments. Three replicate tanks (*n* = 3) per treatment, each containing five juvenile *D. setosum*, served as independent experimental units. This resulted in 90 specimens per seasonal trial (3 replicates × 5 specimens × 6 salinity treatments) and 180 specimens total across both seasons. 

Natural seawater with salinities of 33.68 ± 0.17 in winter and 29.40 ± 0.25 in spring were used for the experiments. The seawater was filtered through a 5 µm filter prior to usage. Six salinity gradients (20, 24, 28, 32, 36, and 40) were established, each with three replicate groups containing five juvenile *D. setosum*.

After measuring test diameter and wet weight, the juveniles were placed in the experimental tanks. Each tank was equipped with an air stone to maintain adequate aeration and ensure dissolved oxygen (DO) saturation. Salinity was adjusted by adding freshwater or sea salt (Sea-Salt Aquarium Technology Co., Ltd., Qingdao, China), with daily adjustments not exceeding 4 units. Target salinity levels were achieved within three days. The juveniles were then cultured for an additional nine days, during which they were fed sufficient benthic diatoms and fresh *Sargassum*. Every two days, two-thirds of the seawater was replaced with water of the corresponding salinity, and residual food and feces were removed via siphoning.

### 2.3. Measurement of Growth and Eco-Physiological Parameters

Following the aquacultural period, juveniles from each tank were transferred to 18 brown 3 L glass jars containing seawater adjusted to their respective salinity levels. Three blank jars filled with seawater only were prepared concurrently as controls. All jars were sealed under water to exclude air bubbles and placed in 150 L temperature-controlled tanks. Water temperature was maintained at 12.06 ± 0.31 °C during winter and 26.25 ± 0.68 °C in spring. The experiment was limited to 8 h to maintain dissolved oxygen (DO) levels in experimental groups at ≥50% of control values, thereby preventing respiration inhibition from oxygen depletion [26]. The seawater in each jar was then gently mixed, and DO was measured using a multi-parameter water quality analyzer (YSI 6920, Yellow Springs, OH, USA). Ammonia nitrogen (NH_4_-N) concentrations were determined from 50 mL water samples according to Chinese Standard GB/T 12763.4-2007 [27]. The atomic O:N ratio determined the primary oxidized energetic substrate, which was calculated using oxygen consumption (OCR) and ammonia excretion (AER) rate values. This calculation follows the equation described by Flores et al. (2008) [28,29]. Fecal samples were collected, rinsed with distilled water to remove salts, oven-dried at 60 °C to constant weight, and weighed to 0.001 g.

Survival rate (*SR*, %), specific growth rate of test diameter (*SGR_D_*, % d^−1^), specific growth rate of wet weight (*SGR_W_*, % d^−1^), oxygen consumption rate (*OCR*, mg·g^−1^·h^−1^), ammonium excretion rate (*AER*, mg·g^−1^·h^−1^), atomic O:N ratio, and fecal production rate (*FPR*, mg·g^−1^·h^−1^) were estimated as follows:*SR* = 100 × *Nt*/*Ni**SGR_W_* = (ln *W_t_* − ln *W*_0_)/*t*_1_ × 100%*SGR_D_* = (ln *D_t_* − ln *D*_0_)/*t*_1_ × 100%*E* = |*C_t_* − *C*_0_| × *V/W_t_* × *t*_2_*O:N* = atomic weight (NH_3_^+^)/atomic weight (O_2_) × [*OCR*]/[*AER*]*FPR* = *F*/*W*_t_ × *t*_2_
where *Nt* is the number of surviving juvenile *D. setosum* at day *t*; *Ni* is the initial number of surviving sea urchins; *W_t_* and *W*_0_ are final and initial wet weights (g); *D_t_* and *D*_0_ are final and initial test diameters (mm); *t*_1_ represents the aquacultural period (d); *E* is OCR or AER; *C_t_* and *C*_0_ are DO or NH_4_-N concentrations (mg L^−1^) in experimental and control groups; *OCR* is oxygen consumption rate; *AER* is total ammonia excretion rate; *V* is the volume of the jar corrected by the volume of juvenile *D. setosum* (L); *t*_2_ is the time of the respiratory metabolism experiments (h); and *F* is the dry weight of feces (mg).

### 2.4. Data Processing

Data were presented as mean ± standard deviation (mean ± SD, *n* = 3). Statistical analysis was conducted using IBM SPSS Statistics for Windows, version 24.0 (IBM Corp., Armonk, NY, USA). Data obtained from different salinity treatments within the same season were subjected to one-way ANOVA, followed by post hoc multiple comparisons with the Student–Newman–Keuls (SNK) test, and comparisons of data from the same salinity treatment between seasons were performed using Student’s *t*-test. Normality and homogeneity of variance were verified before analysis. The level of statistical significance was set at *p* < 0.05.

## 3. Results

### 3.1. Survival

The effects of salinity on the survival rate of juvenile *D. setosum* is shown in Figure 1. Salinity exhibited an extremely significant effect on survival during winter (one-way ANOVA, *p* < 0.001), whereas no mortality was observed across all salinity treatments in spring.

During winter, juvenile *D. setosum* maintained relatively high survival rates (86.7–100%) within the optimal salinity range of 28–36. Specifically, 100% survival was observed at salinities of 32 and 36, whereas the salinity of 28 resulted in a survival rate of 86.67 ± 11.55%. Extreme salinity conditions (20, 24, and 40) significantly reduced juvenile survival. The lowest survival rates were recorded at salinities of 20 (13.33 ± 11.55%) and 24 (26.67 ± 11.55%), with no significant difference between these two groups. Although survival at the salinity of 40 (60.00 ± 20.00%) remained significantly higher than at lower salinities of 20 and 24, it was substantially reduced compared to the optimal salinity range.

A *t*-test analysis showed significant seasonal differences in survival rates for juvenile *D. setosum* at salinities of 20, 24, and 40 (*t*-test, *p* = 0.006, *p* = 0.008, and *p* = 0.026, respectively), with no significant variation observed for other salinity groups (*t*-test, all *p* > 0.05).

### 3.2. Growth

#### 3.2.1. Weight-Specific Growth Rate

Figure 2 illustrates the effect of salinity on the weight-specific growth rate (SGR_W_) of juvenile *D. setosum*. During winter, salinity exerted an extremely significant effect on SGR_W_ (one-way ANOVA, *p* < 0.001). The highest SGR_W_ were observed at salinities of 32 (2.94 ± 0.88%·d^−1^) and 36 (2.19 ± 0.52% d^−1^), which were significantly higher than those at salinities of 20 (−4.50 ± 1.45% d^−1^) and 24 (−0.12 ± 0.14% d^−1^). Notably, juveniles exhibited negative growth within the salinity range of 20–24. No significant differences were detected among other groups. In contrast to winter results, no significant salinity-dependent variations in SGR_W_ were observed during spring (one-way ANOVA, *p* = 0.61), although growth rates showed a tendency to be higher at a salinity of 36 (2.21 ± 1.26% d^−1^) than at 20 (0.81 ± 0.04% d^−1^).

For consistent salinity treatments, a *t*-test analysis revealed significant seasonal differences in SGR_W_ for juvenile *D. setosum* at salinities of 20 and 24 (*t*-test, *p* = 0.035, *p* = 0.023, respectively), while no significant differences were detected between seasons for other salinity groups (*t*-test, all *p* > 0.05).

#### 3.2.2. Test Diameter-Specific Growth Rate

Figure 3 illustrates the effects of salinity on test diameter-specific growth rates (SGR_D_) in juvenile *D. setosum*. Salinity showed highly significant effects on SGR_D_ in both winter and spring (one-way ANOVA, *p* < 0.001, *p* = 0.006, respectively). During winter, juveniles exhibited negative SGR_D_ values at lower salinities (20: −0.74 ± 0.20% d^−1^; 24: −0.28 ± 0.04% d^−1^). Maximum growth rates occurred at the salinity of 32 (1.40 ± 0.37% d^−1^), showing no significant difference from the salinity of 36 (1.12 ± 0.51% d^−1^) but significantly higher than all other salinity treatments. Notably, the SGR_D_ at the salinity of 36 was significantly higher than that at the salinity range of 20–24, while juveniles at salinities of 28 and 40 exhibited significantly higher SGR_D_ (0.37 ± 0.36% d^−1^ and 0.48 ± 0.28% d^−1^, respectively) compared to that at the salinity of 20, while no significant differences were detected among the remaining groups. During spring, juveniles exhibited the highest SGR_D_ at the salinity of 36 (1.02 ± 0.27% d^−1^), significantly exceeding growth rates at salinities of 20 (0.16 ± 0.13% d^−1^), 24 (0.23 ± 0.18% d^−1^), and 40 (0.38 ± 0.11% d^−1^). No significant differences were observed among the remaining groups.

A *t*-test analysis revealed significant seasonal differences in SGR_D_ for juvenile *D. setosum* at salinities of 20 and 24 (*t*-test, *p* = 0.032 and *p* = 0.030, respectively). However, no significant seasonal variations were observed in other salinity groups (*t*-test, all *p* > 0.05).

### 3.3. Eco-Physiological Parameters

#### 3.3.1. Oxygen Consumption Rate

Figure 4 illustrates the effects of salinity on the oxygen consumption rate (OCR) of juvenile *D. setosum*. One-way ANOVA revealed highly significant salinity-dependent variations in OCR during both winter and spring seasons (one-way ANOVA, both *p* < 0.001). During winter, juveniles at the salinity of 32 exhibited peak OCR values (0.290 ± 0.006 mg·g^−1^·h^−1^), which did not differ significantly from those at salinities of 28 (0.258 ± 0.033 mg·g^−1^·h^−1^) and 36 (0.268 ± 0.063 mg·g^−1^·h^−1^). However, these values were significantly higher than those observed at the salinity of 40 (0.054 ± 0.019 mg·g^−1^·h^−1^). During spring, juveniles maintained relatively high OCR levels across the salinity range of 28–40 (0.081–0.100 mg·g^−1^·h^−1^). Within this range, peak OCR occurred at the salinity of 36 (0.096 ± 0.004 mg·g^−1^·h^−1^), significantly exceeding the values observed at salinities of 20 (0.039 ± 0.001 mg·g^−1^·h^−1^) and 24 (0.050 ± 0.001 mg·g^−1^·h^−1^). Notably, juveniles at the salinity of 20 exhibited the lowest OCR values, which differed significantly from all other salinity groups, while no significant differences were detected among the remaining groups.

For consistent salinity treatments, a *t*-test analysis revealed significant seasonal differences in OCR for juvenile *D. setosum* at salinities of 28, 32, and 36 (*t*-test, *p* = 0.017, *p* < 0.001, and *p* = 0.018, respectively). No significant seasonal differences were observed at other salinities (*t*-test, all *p* > 0.05).

#### 3.3.2. Ammonia Excretion Rate

Figure 5 demonstrates the effects of salinity on the ammonia excretion rate (AER) of juvenile *D. setosum*. Salinity significantly affected AER during both winter and spring seasons (one-way ANOVA, both *p* < 0.001). During winter, juveniles at the salinity of 40 exhibited peak AER values (0.037 ± 0.002 mg·g^−1^·h^−1^), which were significantly higher than those measured at salinities of 28 (0.017 ± 0.007 mg·g^−1^·h^−1^), 32 (0.010 ± 0.001 mg·g^−1^·h^−1^), and 36 (0.011 ± 0.004 mg·g^−1^·h^−1^). No significant differences in AER were observed among salinities of 28, 32, and 36. During spring, juveniles maintained consistently low AER levels (0.002–0.004 mg·g^−1^·h^−1^) across the salinity range of 24–40. Within this range, the lowest AER occurred at the salinity of 36 (0.002 ± 0.000 mg·g^−1^·h^−1^). In contrast, the salinity of 20 exhibited significantly higher AER (0.006 ± 0.001 mg·g^−1^·h^−1^) compared to all other treatments.

For consistent salinity treatments, a *t*-test analysis revealed significant seasonal differences in AER for juvenile *D. setosum* at salinities of 32, 36, and 40 (*t*-test, *p* = 0.002, *p* = 0.024, *p* < 0.001, respectively). No significant seasonal differences were observed among other salinity groups (*t*-test, all *p* > 0.05).

#### 3.3.3. Atomic O:N Ratio

Figure 6 illustrates the effects of salinity on the atomic O:N ratio of juvenile *D. setosum*. One-way ANOVA revealed that salinity exerted highly significant effects on the atomic O:N ratio during both winter and spring seasons (one-way ANOVA, both *p* < 0.001). During winter, juveniles at the salinity of 32 exhibited the highest atomic O:N ratio (29.34 ± 3.64), which was significantly greater than those at salinities of 28 (17.30 ± 4.16) and 40 (1.45 ± 0.51), but statistically similar to the salinity of 36 (24.62 ± 3.03). No significant difference was observed between salinities of 28 and 40. During spring, juveniles at the salinity of 36 exhibited the highest atomic O:N ratio (49.76 ± 3.75), which was significantly greater than all other groups. While no significant differences were observed at salinities of 28, 32, and 40 (21.51–35.26), these values were significantly elevated compared to the salinities of 20 (7.13 ± 1.12) and 24 (14.96 ± 1.67). Additionally, the atomic O:N ratio at the salinity of 24 was significantly higher than that at the salinity of 20.

For consistent salinity treatments, a *t*-test analysis revealed significant seasonal differences in the atomic O:N ratio for juvenile *D. setosum* at salinities of 36 and 40 (*t*-test, *p* = 0.002, *p* = 0.002, respectively). No significant variations were observed among other salinity groups (*t*-test, all *p* > 0.05).

### 3.4. Fecal Production Rate

Figure 7 illustrates the effects of salinity on the fecal production rate (FPR) of juvenile *D. setosum*. While salinity exerted highly significant effects on FPR during winter (one-way ANOVA, *p* < 0.001), no significant effects were observed in spring (one-way ANOVA, *p* = 0.38). During winter, juveniles at the salinity of 28 exhibited the highest FPR (13.10 ± 1.64 mg·g^−1^·h^−1^), significantly exceeding all other groups. The second highest FPR occurred at the salinity of 40 (8.34 ± 0.81 mg·g^−1^·h^−1^), which was significantly higher than salinities of 20 (2.14 ± 0.26 mg·g^−1^·h^−1^), 24 (2.70 ± 0.78 mg·g^−1^·h^−1^), and 32 (5.46 ± 0.24 mg·g^−1^·h^−1^). While no significant difference was observed between salinities of 32 and 36 (6.06 ± 2.49 mg·g^−1^·h^−1^), both were significantly higher than the salinities of 20 and 24. There were no significant differences among other groups (all *p* > 0.05). During spring, FPRs remained consistently low across all salinities (0.61–4.40 mg·g^−1^·h^−1^). Juveniles at the salinity of 28 exhibited the highest FPR (2.80 ± 1.60 mg·g^−1^·h^−1^), while those at the salinity of 20 showed the lowest values (1.10 ± 0.49 mg·g^−1^·h^−1^).

For consistent salinity treatments, a *t*-test analysis revealed significant seasonal differences in FPR for juvenile *D. setosum* at salinities of 20, 28, 32, and 40 (*t*-test, *p* = 0.032, *p* = 0.001, *p* < 0.001, *p* < 0.001, respectively). The differences at salinities of 28, 32, and 40 reached highly significant levels (*t*-test, all *p* < 0.001), while no significant seasonal variations were observed at other salinities (*t*-test, all *p* > 0.05).

## 4. Discussion

Salinity represents a critical environmental factor influencing survival, growth, and physiological activities in marine invertebrates [30,31,32,33,34,35,36]. To minimize osmotic shock, we implemented a conservative gradient acclimation protocol, gradually transitioning salinity from ambient seawater conditions to target levels at a rate of 3–4 per day [23,37]. Through rigorous comparative analysis of triplicate groups per salinity treatment, we quantified the survival rates, growth performance, and eco-physiological parameters of juvenile *D. setosum*, thereby determining the optimal aquaculture salinity range to be 32–36. These findings align with research by Sarifudin et al. [23], which demonstrated that *D. setosum* exhibits optimal reproductive performance at the salinity of 31, with fertilization rates, larval survival, and growth all peaking at this level. However, when salinity falls below 28 or exceeds 37, both reproductive capacity and larval development become significantly inhibited. Therefore, we conclude that the suitable salinity range for artificial breeding and aquaculture of *D. setosum* is 28–36.

This study demonstrated that juvenile *D. setosum* could survive normally across all seasons within a salinity range of 28–36. However, during winter, their survival rates at salinities of 20 and 24 were low, indicating weaker tolerance to extreme salinity (≤24) under low-temperature conditions. These results suggest combined effects of salinity and temperature on normal growth and development of the juveniles. Generally, temperature and salinity exert antagonistic effects during early development in echinoderms. For instance, temperature elevation within optimal ranges accelerates developmental rates [38,39,40,41,42,43], while salinity reduction delays development [41,44,45,46,47,48,49]. Although research on combined temperature–salinity effects in echinoderm development remains limited [40], previous studies indicate that while low salinity delays development at optimal temperatures, such negative impacts are diminished when temperature increases remain within optimal ranges [45,49,50,51].

Echinoderms such as *D. setosum* mainly maintain osmotic balance through intracellular isotonic regulation and ionic regulation [19,20,22]. Our experimental results demonstrated that salinity highly significantly affected both specific growth rates in weight (SGR_W_) and diameter (SGR_D_) of juvenile *D. setosum* during winter, while no significant differences were observed during spring. This further confirms the combined effects of salinity and temperature. The juveniles maintained positive growth within a salinity range of 28–40, with optimal growth performance observed at salinities of 32 and 36. In contrast, salinities of 20 and 24 resulted in significantly negative growth rates. This phenomenon likely reflects energetic reallocation toward basal metabolic maintenance (particularly osmoregulation and acid–base balance) under combined low-temperature and hyposaline stress, thereby suppressing test growth. The observed negative values may additionally indicate test thinning or dissolution processes, though mechanistic validation requires further investigation. These findings indicate that the optimal salinity range for juvenile *D. setosum* growth is 32–36, which is similar to the salinity tolerance range of commercially valuable sea urchins such as *S. droebachiensis*, *S. intermedius*, *Mesocentrotus nudus*, and *H. pulcherrimus* [9,52,53,54].

Oxygen consumption rate (OCR) and ammonia excretion rate (AER) are important indicators of energy metabolism in echinoderms [22,55,56]. The oxygen uptake capacity of marine organisms is closely related to their tissue oxygen demand. Higher metabolic intensity corresponds to a greater OCR value, making OCR a key indicator of energy metabolism in echinoderms [22,55,56,57,58]. Juvenile *D. setosum* maintained elevated OCR within suitable salinity ranges (with salinities of 28–36 in winter and 28–40 in spring), positively correlating with enhanced growth rates during these periods. Notably, winter OCR levels at salinities of 28–36 were significantly elevated compared to spring measurements. This physiological difference likely reflects the heightened metabolic demands of juvenile *D. setosum*, as developing organisms require increased nutrient allocation to support tissue formation and organogenesis, resulting in elevated metabolic activity. However, the depressed OCR observed at the salinity of 40 during winter indicated that high salinity at a low temperature had an inhibitory effect on respiratory metabolism.

The AER of echinoderms is closely linked to their protein and amino acid metabolism. Therefore, AER serves as a key indicator of nitrogen metabolism, offering an objective metric for evaluating the nutritional state in echinoderms [22]. In this study, juvenile *Diadema setosum* displayed minimal AER at optimal salinities (32 in winter, 36 in spring), indicating decreased protein catabolism as a metabolic substrate under these conditions. However, under salinity stress conditions (with salinities of 40 in winter and 20 in spring), AER increased significantly, indicating that juvenile *D. setosum* responded to osmotic pressure stress by enhancing nitrogen metabolism. These findings align with documented physiological adaptations to environmental stressors in other sea urchins. For instance, *Paracentrotus lividus* [37] and *S. droebachiensis* [38] exhibit similar metabolic adjustments under ocean acidification (OA)—upregulating ammonia excretion or protein metabolism to maintain homeostasis, ultimately reducing O:N ratios. In conclusion, these findings provide a theoretical basis for studying the physiological adaptation mechanisms of echinoderms such as sea urchins under environmental stress (salinity or OA), especially the establishment of nitrogen metabolism regulation mechanisms.

The atomic O:N ratio is an important indicator for measuring the metabolic process of marine invertebrates. Generally, an atomic O:N ratio greater than 30 indicates that the primary metabolic substrate is carbohydrates or lipids, while a ratio less than 10 suggests that proteins are the main metabolic substrate [55,56,59,60]. In this study, the atomic O:N ratio of the juveniles varied between 1 and 50 across different salinities. The atomic O:N ratio remained above 25 at salinities of 32–36, peaking at 49.76 (salinity of 36 during spring), indicating that carbon-based substrates primarily served as the catabolic substrate for juvenile *D. setosum*. However, under extreme salinity conditions (<24 or >40), the ratio dropped sharply (e.g., to 1.45 at the salinity of 40 during winter), suggesting a shift toward protein catabolism. These metabolic adaptations indicate that under environmental stress (e.g., extreme salinity events), increasing dietary protein content may be necessary to compensate for elevated protein catabolism and maintain normal growth in *D. setosum* aquaculture—a strategy potentially applicable to other tropical sea urchin species.

The growth and metabolism of sea urchins are influenced by various factors such as water temperature, salinity, and size [27,61,62,63,64,65,66]. For instance, the growth rate of *S. intermedius* gradually increases with rising temperature, peaking at the optimal water temperature. Beyond this range, the rate declines steadily. Additionally, larger test diameters in *S. intermedius* correlate with lower optimal growth temperatures and reduced tolerance limit temperatures [65]. Within the suitable temperature range, the AER of both *H. pulcherrimus* [64] and *Anthocidaris crassispina* [61] initially increased and then decreased with rising water temperature, while the OCR increased consistently with temperature. The above results are quite different from those in this study, which may be related to the difference in the size of the juveniles during winter and spring. At present, there have been relatively systematic studies on the influence of temperature on the growth and metabolism of *S. intermedius* of different sizes [66]. However, the synergistic mechanism of temperature and size on the growth and metabolism of *D. setosum* still requires further exploration.

This study found that large-sized juvenile *D. setosum* in spring exhibit higher efficiency in carbon and nitrogen resource assimilation. By reducing their OCR, they allocate more carbon sources toward biomass accumulation. Simultaneously, they significantly decrease nitrogen excretion losses, thereby improving nitrogen utilization efficiency. This metabolic pattern resembles that observed in *S. intermedius* [67]. *S. intermedius* has the highest oxygen consumption rate and the lowest ammonia excretion rate at salinity of 30 treatment, which indicates that the metabolic activity of *S. intermedius* is relatively strong at a salinity of around 30 [67].

The FPR of sea urchins reflects their feeding efficiency, and this indicator is closely related to factors such as the environmental conditions and size [68,69]. This study found that at the same salinity, the FPR of juvenile *D. setosum* in winter was significantly higher than that in spring. This seasonal difference reflects the disparity in feeding efficiency among individuals of different sizes. Meanwhile, under the suitable salinities of 32–36, the FPR of the juveniles is relatively high, reflecting a higher food intake efficiency. Overall, under low-salinity stress conditions (20–24), juvenile *D. setosum* devote more energy to osmotic regulation, which leads to a reduced feeding rate and restricted growth.

## 5. Conclusions

This study demonstrates that salinity and temperature exert significant combined effects on survival rates, growth performance, and eco-physiological parameters of juvenile *D. setosum*. Within optimal temperature ranges, low-salinity impacts are mitigated; however, temperature deviations from optima amplify salinity stress, potentially delaying or inhibiting normal development. The optimal salinity range of 32–36 supported maximal growth rates and peak energy metabolism efficiency. Corresponding declines occurred when salinity fell below 32 or exceeded 36. Collectively, these findings provide a theoretical foundation for the artificial breeding and aquaculture of tropical sea urchins.

## Figures and Tables

**Figure 1 animals-15-02462-f001:**
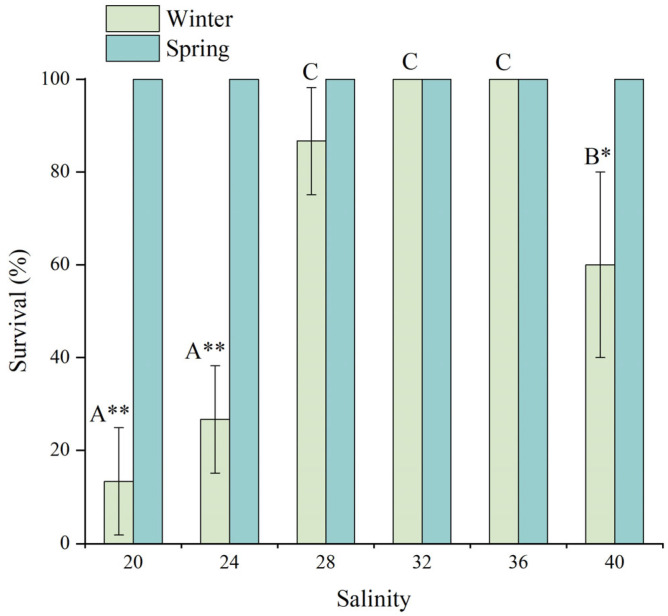
The effects of salinity on the survival rate of juvenile *Diadema setosum* during winter and spring. The data represent means ± SD (*n* = 3; derived from three replicate groups per treatment, each comprising five juveniles). Different uppercase letters (A, B, C) indicate significant differences among salinity treatments within the winter group (*p* < 0.05). Bars that share a common letter are not significantly different from each other. Asterisks mark significant differences between winter and spring at identical salinities (* *p* < 0.05, ** *p* < 0.01).

**Figure 2 animals-15-02462-f002:**
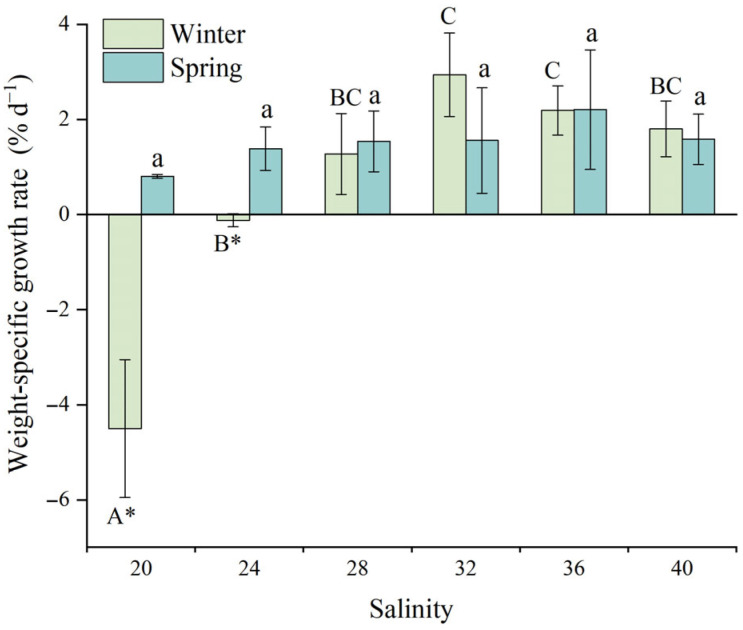
The effects of salinity on the weight-specific growth rate of juvenile *Diadema setosum* during winter and spring. The data represent means ± SD (*n* = 3; derived from three replicate groups per treatment, each comprising five juveniles). Different uppercase letters (A, B, C) indicate significant differences among salinity treatments within the winter group (*p* < 0.05), while lowercase letters (a) indicate no significant differences within the spring group (*p* > 0.05). Bars that share a common letter are not significantly different from each other. Asterisk marks significant differences between winter and spring at identical salinities (* *p* < 0.05).

**Figure 3 animals-15-02462-f003:**
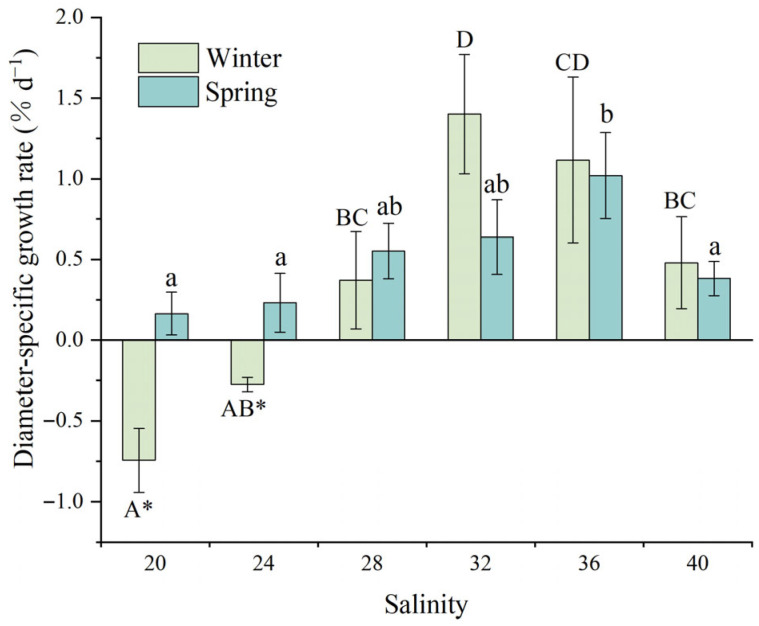
The effects of salinity on the test diameter-specific growth rate of juvenile *Diadema setosum* during winter and spring. The data represent means ± SD (*n* = 3; derived from three replicate groups per treatment, each comprising five juveniles). Different uppercase letters (A, B, C, D) indicate significant differences among salinity treatments within the winter group, while different lowercase letters (a, b) indicate significant differences within the spring group (*p* < 0.05). Bars that share a common letter are not significantly different from each other. Asterisk marks significant differences between winter and spring at identical salinities (* *p* < 0.05).

**Figure 4 animals-15-02462-f004:**
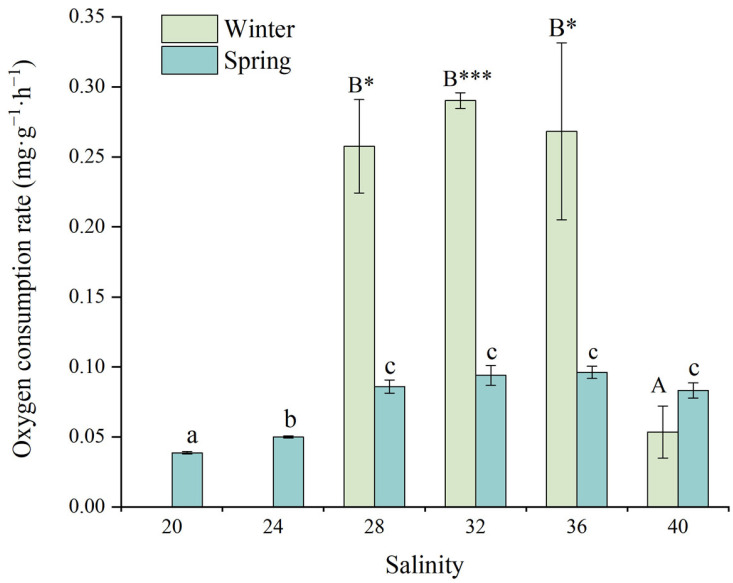
The effects of salinity on the oxygen consumption rate of juvenile *Diadema setosum* during winter and spring. Note: In winter, the salinities of 20 and 24 did not obtain the oxygen consumption data due to the high mortality of juvenile *Diadema setosum*. The data represent means ± SD (*n* = 3; derived from three replicate groups per treatment, each comprising five juveniles). Different uppercase letters (A, B) indicate significant differences among salinity treatments within the winter group, while different lowercase letters (a, b, c) indicate significant differences within the spring group (*p* < 0.05). Bars that share a common letter are not significantly different from each other. Asterisks mark significant differences between winter and spring at identical salinities (* *p* < 0.05, *** *p* < 0.001).

**Figure 5 animals-15-02462-f005:**
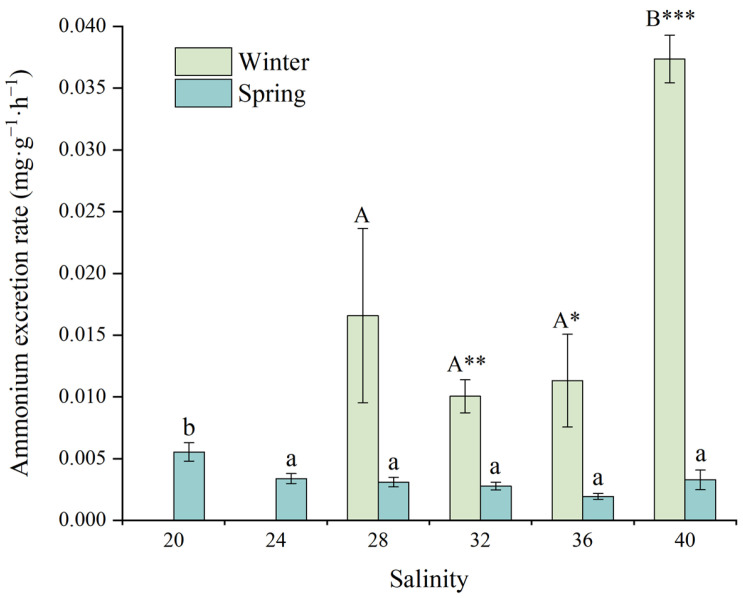
The effects of salinity on the ammonium excretion rate of juvenile *Diadema setosum* during winter and spring. Note: In winter, the salinities of 20 and 24 did not obtain the ammonium excretion data due to the high mortality of juvenile *Diadema setosum*. The data represent means ± SD (*n* = 3; derived from three replicate groups per treatment, each comprising five juveniles). Different uppercase letters (A, B) indicate significant differences among salinity treatments within the winter group, while different lowercase letters (a, b) indicate significant differences within the spring group (*p* < 0.05). Bars that share a common letter are not significantly different from each other. Asterisks mark significant differences between winter and spring at identical salinities (* *p* < 0.05, ** *p* < 0.01, *** *p* < 0.001).

**Figure 6 animals-15-02462-f006:**
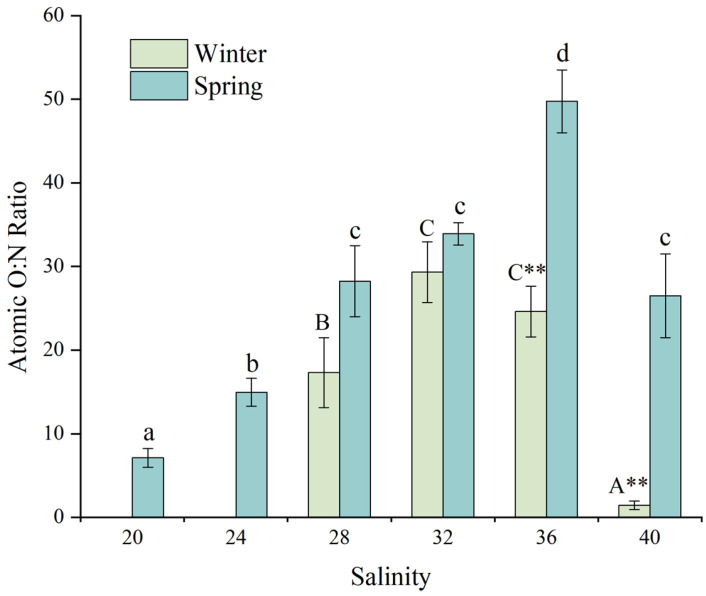
The effects of salinity on the atomic O:N ratio of juvenile *Diadema setosum* during winter and spring. Note: In winter, the salinities of 20 and 24 did not obtain the atomic O:N ratio due to the high mortality of juvenile *Diadema setosum*. The data represent means ± SD (*n* = 3; derived from three replicate groups per treatment, each comprising five juveniles). Different uppercase letters (A, B, C) indicate significant differences among salinity treatments within the winter group, while different lowercase letters (a, b, c, d) indicate significant differences within the spring group (*p* < 0.05). Bars that share a common letter are not significantly different from each other. Asterisk marks significant differences between winter and spring at identical salinities (** *p* < 0.01).

**Figure 7 animals-15-02462-f007:**
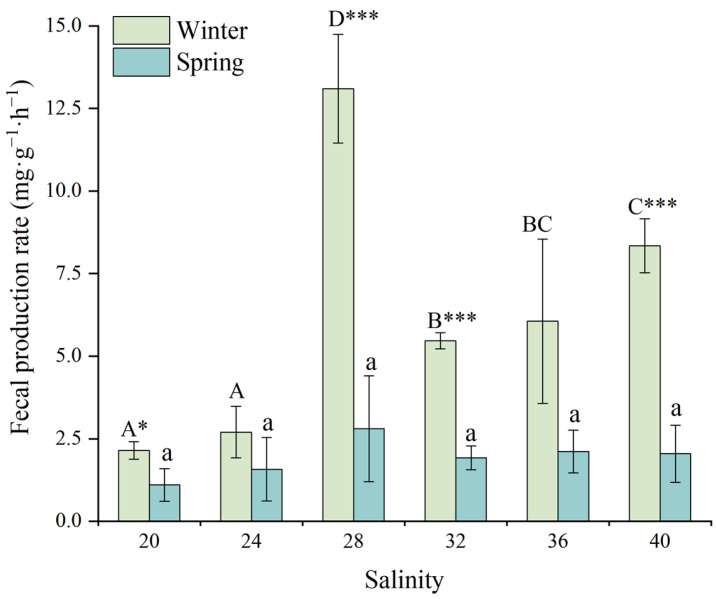
The effects of salinity on the fecal production rate of juvenile *Diadema setosum* during winter and spring. The data represent means ± SD (*n* = 3; derived from three replicate groups per treatment, each comprising five juveniles). Different uppercase letters (A, B, C, D) indicate significant differences among salinity treatments within the winter group, while lowercase letters (a) indicate no significant differences within the spring group (*p* > 0.05). Bars that share a common letter are not significantly different from each other. Asterisks mark significant differences between winter and spring at identical salinities (* *p* < 0.05, *** *p* < 0.001).

## Data Availability

The original contributions presented in this study are included in the article/Appendix A. Further inquiries can be directed to the corresponding author.

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
