# Peer review of "The Effects of Salinity on the Survival, Growth, and Eco-Physiological Parameters of Juvenile Sea Urchin Diadema setosum"

_animals, 2025, doi:10.3390/ani15162462_

Round 1
Reviewer 1 Report
Comments and Suggestions for Authors
The manuscript studied of the physiology of the sea urchin species Diadema setosum. Given the commercial and ecological importance of this species, the study undoubtedly holds both scientific and practical interest. The data presented are novel and relevant. However, the manuscript is written in a rather careless and confusing manner. It requires substantial revision and supplementation in several sections.
General comments:
Diadema setosum is an extremely popular species for various types of research. In my opinion, the Introduction section needs to be supplemented with known facts about the biology of this species (for example, the range of natural temperatures and salinities). This would provide a stronger justification for the salinity levels chosen in the experiments and emphasize the relevance of the study's objectives.
The Materials and Methods section needs major revision and supplementation. There is no information regarding the origin of the specimens used for the experimental work. Detailed data about the control groups, which were maintained under native salinity conditions but in the same facilities, are lacking. The description of the methods for primary data collection and analysis is insufficient for a complete understanding. More specific comments are provided below.
The Discussion section pays insufficient attention to temperature and its effects, despite the significant influence of temperature being directly evident from the presented data. Since it remains unclear which specific data were analyzed (average per aquarium or individual specimens), it is difficult to assess the appropriateness of the statistical methods used and the correctness of the interpretation of the results. This point requires clarification in both the Materials and Methods section and the Results section. It may be more appropriate to present the data in terms of the number of individuals (Figure 1) rather than percentages, as well as in proportions (Figures 2 and 3).
The descriptions accompanying the figures lack all necessary details and should be expanded. What do the uppercase and lowercase letters (ABCD) on the graphs represent? This should be included in the figure legends.
The figures adequately represent the results obtained; however, for such experimental work, it is essential to include illustrations that clarify the Materials and Methods section. A schematic representation of the experiment would be extremely helpful, as well as photographs of the aquariums and their setup.
Detailed questions:
1) Section 2, Item 2.1: Were the animals used in the experiments collected from the wild? What collection methods were employed, and how was the collection organized? In which location were the animals collected?
2) Section 2, Item 2.2: Were there control groups of animals maintained at natural salinity in the same facility, fed the same diet, and subject to the same water exchange regime as the experimental specimens? What was the acclimation period for transferring the animals to different salinities? How was it determined that the adaptation to the experimental conditions was complete?
3) Section 2, point 2.3, Lines 113 and 114: Why do you use the logarithmic values of weight and size parameters?
4) Section 2, point 2.3, Lines 115 and 117: What do the variables t2 mean in these formulas represent?
5) Section 2, point 2.4, Line 126: Why do you use n=3? What data are you comparing? Are you comparing the aquariums as a whole or individual specimens? It remains unclear how the primary data were processed and modified. What exactly constituted the primary data for this study? What specific data were compared using statistical methods?
6) Section 3, point 3.2.2, Figure 3: How did you determine the diameter of the shell? What explanations can you offer for the observed reduction in shell size?
7) Section 3, 3.3.1, 3.3.2..., Figures 4 and 5 and below: Why are there no data for low salinity values?
8) Section 4, Lines 297-299: These lines belong to the Materials and Methods section. Additionally, literature or experimental evidence is needed to demonstrate that the salinity transition did not induce a stress response in the juveniles.
9) Section 4, Line 302: According to the authors' data, low salinity alone is not a key factor, as the juveniles grew under these parameters during the spring period. It is likely that there is a combined effect of salinity and temperature factors, and possibly that smaller individuals (used in winter) are less resilient than larger ones (used in the spring experiments).
10) Section 4, Lines 316-317: This is only shown for winter conditions. However, for spring, the authors do not find significant differences in growth parameters of size or weight. This thesis should be rephrased in accordance with the obtained data.
11)Section Conclusion, the significant differences in physiological parameters related to salinity are shown only for winter. In contrast, for spring, the authors do not find significant differences between experimental animals inhabiting different salinities. This requires at least a discussion of additional factors, primarily temperature and size of the specimens, and a reformulation of the conclusion considering possible joint influences.
My overall conclusion is that the article may be accepted for publication after major revision and supplementation.
Comments on the Quality of English Language
Unfortunately I can't estimat English language quality, so I would recommended consulting other specialists.
Reviewer 2 Report
Comments and Suggestions for Authors
Review for the paper submitted to “Animals”.
Title: The Effects of Salinity on the Survival, Growth and Eco-physiological Parameters of Juvenile Sea Urchin Diadema setosum
Authors: Xuanliang Wang, Jieyu Zhang, Lei You, Yunyong Jin, Zhenhao Lin, Junhao Lin, Jinhui Wu, Zonghe Yu
The authors focused on determining the optimal salinity conditions for cultivating the economically significant sea urchin Diadema setosum in aquaculture settings, examining its effects on juvenile survival, growth, and key physiological responses across different seasons. The authors' study showed that juvenile Diadema setosum demonstrates a distinct preference for moderate salinity conditions for both survival and robust growth. Optimal performance consistently occurred within a specific moderate salinity band. The study revealed a significant shift in metabolic strategy linked to salinity stress.
There are the following implications of the authors' work:
Understanding the precise salinity tolerance and optimal range for Diadema setosum is crucial for designing effective and sustainable aquaculture systems. This knowledge also informs site selection for sea ranching or restocking efforts. Beyond direct cultivation, these findings contribute to predicting how this species might respond to changing oceanic conditions, such as altered salinity patterns resulting from climate change impacts.
Suggestions for improving the paper:
Abstract.
L 31. The authors should avoid using unexplained abbreviations in the abstract.
Introduction.
L 41-43. The authors stated that sea urchins are "prized as a premium marine delicacy" and "valued for their nutrient-rich gonads". They should clarify what specific market prices or economic value ranges constitute "premium". They also should provide examples of the key beneficial unsaturated fatty acids present in the gonads.
The authors should expand the introduction section with relevant citations and information regarding the ecology and aquaculture of sea urchins worldwide. They should consider the following papers:
James, P., Siikavuopio, S.I., Mortensen, A. 2015. Sea urchin aquaculture in Norway. In: Brown N.P., Eddy, S.D. (Eds.) Echinoderm Aquaculture. John Wiley & Sons, Inc., Hoboken, New Jersey, pp 147–173.
Angwin, R. E., Hentschel, B.T., Anderson, T. W. 2022. Gonad enhancement of the purple sea urchin, Strongylocentrotus purpuratus, collected from barren grounds and fed prepared diets and kelp. Aquaculture International, 30, 1353-1367.
Dvoretsky, A.G., Dvoretsky, V.G. 2020. Aquaculture of green sea urchin in the Barents Sea: A brief review of Russian studies. Reviews in Aquaculture, 12, 2080-2090.
Rogers-Bennett, L., Okamoto, D. 2020. Mesocentrotus franciscanus and Strongylocentrotus purpuratus. In: Developments in Aquaculture and Fisheries Science (Vol. 43, pp. 593-608). Elsevier.
Dvoretsky, A.G., Dvoretsky, V.G. 2024. Distribution patterns and biological aspects of Strongylocentrotus droebachiensis (Echinoidea: Echinoida) in Russian waters of the Barents Sea: Implications for commercial exploration. Reviews in Fish Biology and Fisheries, 34, 1215–1229
L 51-52. The authors should report the primary causes of this natural resource depletion. It would be useful to provide any data on the magnitude of documented population declines for this species.
L 61-64. The authors should update the text with information for Strongylocentrotus droebachiensis. See, Dvoretsky and Dvoretsky 2020, page 2083.
L 65-66. The authors cited research indicating D. setosum has "narrow salinity tolerance (optimal at 31)" with impairment below 28 or above 37. They should clarify the specific life stages and physiological processes to which this cited tolerance range applies.
Materials and Methods.
L 86. How many sea urchins were used in the experiments? The total sample size should be reported here.
L 91-93. The authors stated that the juveniles were fed "sufficient benthic diatoms and fresh Sargassum". They should report the specific daily ration (percentage of body weight or absolute amount per individual/group) and the relative proportion of diatoms to Sargassum to ensure feeding was standardized and ad libitum. They should also explain the rationale for choosing this relatively short duration (9 days) for assessing growth and physiological parameters.
L 103. The authors stated that the experiment ensured DO "did not fall below 50% of the controls". They should clarify the rationale for choosing this specific 50% depletion threshold.
L 130. The authors stated that data were presented as "mean ± standard deviation (mean ± SD, n = 3)" and analyzed with one-way ANOVA. They should clarify what the "n = 3" represents (the 3 replicate tanks, or the 15 individuals per treatment pooled by tank?).
Results.
L 159. The authors should delete "for each sea urchin species".
Figures 2-7. The authors should explain the meaning of upper and lower case letters.
Discussion.
L 298-300. This sentence is redundant and should be deleted.
L 319. The authors noted "negative growth rates" at salinities of 20 and 24 in winter. They should explain the possible physiological mechanism causing this shrinkage.
L 321-322. The authors should include the species Strongylocentrotus droebachiensis in this list according to information presented in (Dvoretsky and Dvoretsky 2020).
L 359. The authors highlighted the O:N ratio shift indicating a metabolic substrate switch under stress. They should discuss the practical implications of this shift for aquaculture. Does protein catabolism imply higher dietary protein requirements under suboptimal salinity?
The authors identified salinities 32-36 as optimal based on a 9-day experiment. Could longer-term culture at the edges of this range (28, 36, and 40) reveal cumulative negative effects on health, reproduction, or immunity not captured here?
Reference list.
L 420. Diadema setosum should be italicized.
